# Comparison of Phenolic and Volatile Compounds in MD2 Pineapple Peel and Core

**DOI:** 10.3390/foods12112233

**Published:** 2023-06-01

**Authors:** Nur Liyana Nordin, Rabiha Sulaiman, Jamilah Bakar, Mohd Adzahan Noranizan

**Affiliations:** 1Laboratory of Halal Product Science, Halal Products Research Institute, Universiti Putra Malaysia, Putra Infoport, Serdang 43400, Selangor, Malaysia; nliyana.nln@gmail.com (N.L.N.); rabiha@upm.edu.my (R.S.); jamilah@upm.edu.my (J.B.); 2Department of Food Technology, Faculty of Food Science and Technology, Universiti Putra Malaysia, Serdang 43400, Selangor, Malaysia

**Keywords:** MD2, pineapple waste, radical scavenging activity, phenolic compound, volatile compounds

## Abstract

The peel and core discarded from the processing of MD2 pineapple have the potential to be valorized. This study evaluated the functional and volatile compounds in the extracts of MD pineapple peel and core (MD2-PPC). The total soluble solids, pH, titratable acidity, sweetness index, and astringency index were 9.34 °Brix, 4.00, 0.74%, 12.84, and 0.08, respectively, for the peel and 12.00 °Brix, 3.96, 0.32%, 37.66, and 0.03, respectively, for the core. The fat and protein contents of the peel and core were found to be significantly different (*p* < 0.05). The total phenolic (TPC) and flavonoid contents (TFC) were significantly higher in the peel. The peel also showed better antioxidant activity, with a half-maximal inhibitory concentration (IC_50_) of 0.63 mg/mL for DPPH free radical activity compared with the core. The TPC of different phenolic fractions from peel extract was highest in the glycosylated fraction, followed by the esterified, insoluble-bound, and free phenolic fractions. GC–MS analysis identified 38 compounds in the peel and 23 in the core. The primary volatile compounds were 2-furan carboxaldehyde, 5-(hydroxymethyl), and 2,3-dihydro-3,5-dihydroxy-6-methyl-4H-pyran-4-one (DDMP). The identification of phenolics and volatile compounds provides important insights into the valorization of (MD2-PPC) waste.

## 1. Introduction

The increasing global production of food waste is a consequence of the rising human population and the disparity in the food supply chain [1]. More than 1.3 billion tons of food waste are produced annually [2]. The highest proportion of generated waste is fruit and vegetables (42%), followed by dairy products (26%), cereals and grains (19%); other food waste (13%) [1,3]. Fruit peel wastes account for up to 90–92% of fruit and vegetable processing wastes, with an estimated loss of 55 MMT (million metric tons) [4]. The disposal route for these wastes is landfill, which causes global warming through the release of greenhouse gases and methane [5]. Fruit waste has the potential to be utilized as a food ingredient as it is a low-cost raw material [6]. Given the urgent requirement to produce high-value ingredients from agricultural by-products, researchers are attempting to decrease food waste in compliance with one of the United Nations’ Sustainable Development Goals [7].

Bioactive compounds, such as phenolics, are mainly stored in the coating layer of plants (peel, shell, husk, and seed coat) because of the higher contents of starch, protein, and lipids found in the flesh [8]. The phenolic fractions consist of various forms such as free, soluble conjugate (esterified and glycosylated), and insoluble bound phenolics [9]. Free phenolics are the easiest to extract using chemical solvents, whereas soluble conjugates are esterified to sugar molecules via hydroxyl or carbon–carbon bonds [10]. Insoluble bound phenolics (IBP) are localized within the plant cell wall tissue structure and covalently bound to pectin, cellulose, hemicellulose, and structural proteins [11]. According to one study, apple peel contains more total phenolic (TPC) and flavonoid content (TFC) than apple core and flesh [12]. Montefusco et al. [13] also highlighted that the total flavonoids in pomegranate peels were higher than in the fruit juice. TPC amounts were higher in the peel than the pulp, such as in MD2 pineapple [14]; Smooth Cayenne, Tainung 17, and Tainung 19 pineapples [15]; peach [16]; kaffir lime [17]; orange and banana [18]. According to Zainal Arifin et al. [19], the abundance of phenolics peels indicates the potential of fruit peels as extracts in active and intelligent packaging which could increase the quality and shelf-life of food products.

The global production of pineapples, produced by the Asian and American regions [20], will reach 33 million tons by the year 2029. In Europe, an estimated 1.45 million tons of pineapples were imported, of which only half was used in the form of juice, canned fruits, crystallized fruit, and dehydrated snacks [21]. However, postharvest, processing, storage, and distribution result in a loss of around 1.3 billion metric tons of this fruit [22,23]. In the Southeast Asian region, pineapple waste from canning industries accounts for up to 75% *w*/*w* of the original fruit [24]. The peel made up the most significant percentage of waste, comprising around 30% to 40% *w*/*w* of the original fruit. In contrast, the core and stem were disposed of in a much smaller amount, making up under 5% *w*/*w* of the initial fruit weight. [6]. Pineapple waste parts constitute the crown, core, peel, and stem, which have many industrial applications such as starches, proteins [22], fiber, phenolics [21], and flavorings [20,25,26,27]. The chemical and biochemical properties of pineapple are often categorized based on the cultivar, maturity level, ripeness, and postharvest handling processes [28]. MD2 has been selected as the main exported fruit due to its good preservation properties, improved flavor, bright gold–yellow pulp, low acidity (0.40–0.45%), thinner peel, and longer shelf-life [29,30,31].

Aroma compounds constitute 25% of the food additives industry [32]. Flavors and aromas are for masking unpleasant or bland odors in food products. However, the use of synthetic flavors raises food safety issues. For example, a daily intake of vanillin flavor is 10 mg/kg in the human diet, and an intake of over 75 g/kg, could cause toxicity resulting in 50% of allergic responses [33]. Therefore, natural aroma compounds derived from fruit waste could be alternatives that confer health benefits. Natural aromas and flavors are derived from fruit and vegetable juices, spices, herbs, and bark in the form of essential oils, essence, extracts, distillate, or the products of smoking, fermentation, and heating [34]. Fruit aromas are a complex mixture of volatile compounds influenced by the environmental condition [35]. The compounds utilized as food flavoring additives are formed from a variety of chemicals, including hydrocarbons, alcohols, aldehydes, ketones, acids, esters, and lactones [34]. In pineapples, the major aroma compounds contain esters (35%) and aldehydes [36]. In MD2 pineapple fruit, methyl hexanoate, 3-methylbutanoic acid, methyl-3-hydroxy hexanoate (ester), hexadecanoic acid, and 2-methoxy-4-vinyl phenol are the major aroma compounds [37,38]. As reported by Garcia et al. [20], the pineapple peel and core of unknown variety contained ethyl acetate and isopentyl acetate (ester) as the major aroma compounds. The characterization of aroma compounds in the pulp and core of Smooth Cayenne pineapple identified 44 volatile compounds, of which 2,5-dimethyl-4-hydroxy-3(2H)-furanone (DMHF) was found to be the dominant aroma [39]. A recent review published by George et al. [25] stated that pineapples have unique aroma chemistry profiles influenced by their respective maturity levels and different varieties. The findings involving various aroma compounds derived from various pineapple varieties and waste could lead to applications in the food flavoring and fragrance processing industries.

Therefore, this study aimed to evaluate the different phenolic fractions and the volatile compound profiles of the MD2-PPC variety. As the specific variety and maturity index level influence each of the aroma chemistry characteristics of pineapple, this study focused on the MD2-PPC cultivated in Malaysia with a maturity index of 2. Furthermore, the extraction of different phenolic fractions would identify the presence of bound phenolics in the peel. Volatile profiles were identified to compare each of the aroma compounds in the MD2-PPC variety. The findings from this study may benefit the food industry as a potential natural source of food and flavoring ingredients, such as synthetic antioxidants, and aroma, as well as reducing food waste.

## 2. Materials and Methods

### 2.1. MD2 Pineapple Peel and Core (MD2-PPC)

MD2-PPC with a maturity index of 2, where 25% of the peel base was yellow, were collected from the Department of Agriculture in Serdang, Selangor, Malaysia. The selection of MD2-PPC at a maturity index of 2 was performed in accordance with the method of Mohd Ali et al. [26]. The MD2-PPC were packed into a 5 kg PP woven bag, brought to the laboratory, washed under running water to remove dirt, and stored overnight at −18 °C ± 5 °C until sample preparation. The frozen peels and cores were weighed (250 g) and thawed in a chilled refrigerator (4 °C ± 5 °C) for 60 min before extraction and analysis. For every peel and core, samples were analyzed in triplicate.

### 2.2. Chemicals

Analytical (AR) grade chemical reagents were used in the analyses. Ethanol absolute (C_2_H_6_O) was purchased from John Kollin Corporation (Midlothian, Edinburgh, UK). Petroleum ether 40–60 °C (C_6_H_14_), sodium hydroxide (NaOH), phenolphthalein indicator, butylated hydroxyanisole (BHA), and Folin–Ciocalteu phenol reagent were purchased from Merck KgaA (Darmstadt, Germany). Aluminum chloride (AlCl_3_), sodium carbonate (Na_2_CO_3_) and sodium nitrite (NaNO_2_) were acquired from R & M Chemicals Sdn Bhd (Selangor, Malaysia). Gallic acid (C_7_H_6_O_5_) and quercetin (C_15_H_10_O_7_) were purchased from Sigma-Aldrich (Steinheim, Germany), and 2,2-diphenyl-1-picryl-hydrazyl-hydrate (DPPH) was purchased from Alfa Aesar and Thermo Fischer Scientific (Lancashire, UK).

### 2.3. Preparation of Liquid Extract

Phenolics were extracted from the MD2-PPC using a solid–liquid extraction method as described by Hossain and Rahman [40]. The liquid extract of 250 g from MD2-PPC was collected using a high-speed juice extractor (Cornell Juice Extractor 250–300 W, CJX-SP450 Brown, Malaysia) as shown in Figure 1. To 25 mL of each extract, 150 mL of distilled water was added and mixed using a digital mechanical plate stirrer (120V, IKA Works Incorporation, Staufen, Germany) at 30 °C ± 5 °C for 2 h. Then, the extract was filtered through Whatman® filter paper No. 4 (125 mm). Before analysis, the liquid extract was obtained, transferred into airtight bottles, and stored at −18 °C ± 5 °C.

### 2.4. Composition

#### 2.4.1. Proximate Analysis

##### Moisture Content

Moisture content was performed in accordance with a method established in AOAC [41]. Five grams of each sample was dried at 105 °C in a forced-air oven (Memmert UNB 100, Memmert GmbH + Co. KG, Schwabach, Germany) until a constant weight was reached. The moisture content was calculated based on the weight difference between the initial and final weight after drying, as shown in the calculation below (1):(1)$$\mathrm{Moisture}\;(\%\;w/w)=\frac{\mathrm{Initial}\;\mathrm{weight}-\mathrm{Final}\;\mathrm{weight}}{\mathrm{Initial}\;\mathrm{weight}}\;\times \;100$$

##### Ash Content

Ash content was performed according to the method as described in AOAC [41]. Approximately 3 g of each peel and core samples were incinerated in a muffle furnace at 550 °C for 5 h. The ash content was determined based on the difference in the initial and final weight of the crucibles before and after incineration as shown in the Equation (2). The result was expressed as g/100 g FW (fresh weight).
(2)$$\mathrm{Ash}\;\mathrm{content}\;(g/100\;g)=\frac{\mathrm{Final}\;\mathrm{weight}\;\mathrm{of}\;\mathrm{sample}}{\mathrm{Initial}\;\mathrm{weight}\;\mathrm{sample}}\;\times \;100$$

##### Crude Fat Content

The Soxhlet extraction method was evaluated for the determination of the crude fat content in the peel and core [41]. Briefly, 3 g of each sample was extracted with 90 mL of petroleum ether 40–60 °C (C_6_H_14_) using an automated fat extraction machine (Soxtec^TM^ 2050 Auto Fat Extraction System, FOSS Analytical, Hilleroed, Denmark). Crude fat was determined based on the dry weight of the obtained sample extract. The result was expressed as g/100 g FW.

##### Protein Content

The micro-Kjeldahl method was performed to evaluate the protein contents of the peel and core [41]. Two grams of defatted samples (obtained from the remaining sample after crude fat content) was analyzed using the Automatic Kjeldahl Analyzer for distiller and digestion (Vadopest VAP20, Gerhardt GmbH + Co. KG, Königswinter, Germany). Protein content was determined based on the Equation below (3):(3)Protein (g/100 g fresh weight (FW))=% nitrogen ×conversion factor (6.25)

##### Total Carbohydrate Content

The carbohydrate content of MD2-PPC samples was determined based the study of Matsuo et al. [42], using a different method as shown in Equation (4) below:(4)$$\mathrm{Carbohydrate}\;\mathrm{content}\;\left(\frac{g}{100\;g}\right)=100-\left(\mathrm{moisture}+\mathrm{fat}+\mathrm{ash}+\mathrm{protein}\right)$$

#### 2.4.2. Chemical Composition

##### pH

The pH value of MD2-PPC was measured using a pH meter (Thermo Scientific™ Eutech™ pH 700 Meter, Fischer Scientific, Lougborough, Leicestershire, UK). The probe was standardized before use with pH 4 and pH 7 buffers. Peel and core extracts (10 mL) were placed in a 50 mL beaker and measured using a glass electrode probe. Analysis was performed in triplicate.

##### Titratable Acidity

Titratable acidity (anhydrous % citric acid) was performed in accordance with AOAC [43]. To 10 mL of peel and core extracts, 250 mL of deionized water was added. One milliliter of phenolphthalein indicator (C_20_H_14_O_4_) was added to the mixture. The mixture was then back titrated with 0.1 N of sodium hydroxide (NaOH) standard solution until a faint pink color (the endpoint) was observed. Analysis was performed in triplicate. The percentage of acidity was calculated in accordance with Equation (5) below:(5)$$\mathrm{Acid}\;\left(\mathrm{as}\;\mathrm{anhydrous}\;\%\;\mathrm{citric}\;\mathrm{acid}\right)=\mathrm{Volume}\;\mathrm{of}\;0.1\;N\;\mathrm{NaOH}\;\left(\mathrm{mL}\right)\times \frac{0.64}{10.00}$$

##### Soluble Solids

Total soluble solids determination in MD2-PPC was performed using a Hand-Held Refractometer MASTER-53α, Atago, Japan (range of 0.0% to 53.0%). The temperature of the sample was 20.12 °C. One drop of MD2-PPC was placed on the prism, and the total soluble solids were read and recorded in °Bx. Analysis was performed in triplicate.

##### Sweetness Index (SI) and Astringency Index (AI)

The sweetness index (SI) and astringency index (AI) were measured as described by Wardy et al. [43]. The SI was calculated as the ratio of total soluble solids to titratable acidity, whereas the AI was calculated as the ratio of titratable acidity to soluble solids content, respectively.

### 2.5. Total Phenolic Content (TPC)

TPC of the MD2-PPC extracts was performed according to a method by Hossain and Rahman [40]. First, 200 µL of each extract was transferred into a test tube. Then, 0.2 mL of Folin–Ciocalteu solution was added, and the tube was vortexed. After 4 min, 1 mL of 15% sodium carbonate (Na_2_CO_3_) solution was added to the mixture and it was left to stand in the dark for 2 hours at room temperature. The blank solution was prepared in the same manner, except without the addition of extracts. The absorbance of mixtures was read at 765 nm against the blank using a Thermo Scientific GENESYS^TM^ 10S UV–Vis Spectrophotometer (Waltham, MA, USA). The gallic acid stock solutions were used to generate the standard calibration curve of different gallic acid concentrations (0, 25, 50, 100, 150, and 250 µg/mL). The total phenolic content was measured as milligrams of gallic acid equivalents (GAE)/100 g of FW using Equation (6) below:(6)$$\mathrm{TPC}\;\left(\mathrm{mg}\frac{\mathrm{GAE}}{100\;g}\right)=C_{1}\times \mathrm{DF}\;\times \frac{v}{m}$$
where TPC is the total phenolic content in mg GAE/100 g, C_1_ is the concentration of gallic acid obtained from the standard curve in mg/mL, DF is the dilution factor, *v* is the volume of extract in mL, and *m* is the weight of the plant extract in grams.

### 2.6. Total Flavonoid Content (TFC)

The TFC in MD2-PPC extracts was determined as reported by Zhishen et al. [44]. A mixture of 1 mL of extract, 4 mL of distilled water, and 0.3 mL of 10% sodium nitrite (NaNO_2_) solution was vortexed and allowed to stand for 5 min. Then, 0.3 mL of 10% aluminum chloride (AlCl_3_) was added, followed by the addition of 2 mL of 1 M sodium hydroxide (NaOH). The blank solution was prepared in the same manner, except without the addition of extract. The solutions were vortexed and the absorbance at 510 nm was immediately measured using a Thermo ScientificTM GENESYS^TM^ UV–Vis Spectrophotometer (Waltham, MA, USA). The quercetin stock solutions were used to generate the standard calibration curve of different quercetin concentrations (0, 25, 50, 100, 130, and 160 µg/mL). The TFC was expressed in milligrams of quercetin equivalents (QE)/100 g of FW and calculated using Equation (7) below:(7)$$\mathrm{TFC}\;\left(\mathrm{mg}\frac{\mathrm{QE}}{100\;g}\right)={\;C}_{1}\times \mathrm{DF}\times \frac{v}{m}$$
where TFC is the total flavonoid content in mg QE/100 g, C_1_ is the concentration of quercetin obtained from the standard curve in mg/mL, DF is the dilution factor; *v* is the volume of extract in mL, and *m* is the weight of the plant extract in grams.

### 2.7. Free Radical Scavenging and IC_50_ Assay

The antioxidant activity of MD2-PPC was evaluated by the 2,2-diphenyl-1-picryl-hydrazyl-hydrate (DPPH) free radical scavenging assay of Hossain and Rahman [42] with modifications to the sample preparation. Different concentrations of 25 µL, 50 µL, and 100 µL volumes of each peel and core extract were placed into different test tubes and ethanol was added to a final volume of 100 µL. Ethanol (C_2_H_6_O) was used as the blank sample. Then, different extract concentrations were added with 5 mL of ethanolic DPPH solution (0.1 mM). The mixtures were vortexed and incubated in the dark at 22 °C ± 1 °C for 20 min. The preparation of control sample was completed without the addition of extract. A standard reference of butylated hydroxyanisole (BHA) was prepared in the same manner. Absorbance values at 517 nm were read using a Thermo Scientific^TM^ GENESYSTM 10 Series UV–visible spectrophotometer. The DPPH free radical scavenging assay was measured as described in Equation (8):(8)$$Inhibition\;\left(\%\right)=\left(\frac{Control\;Absorbance-Sample\;Absorbance}{Control\;Absorbance}\right)\times 100$$

The IC_50_ was calculated as the antioxidant concentration of the sample able to inhibit 50% of DPPH activity. The calculation was based on the percentage of radical scavenging activity against a concentration of the sample as described by Li et al. [45].

### 2.8. Sample Preparation for the Extraction of Free, Soluble Conjugate, and Insoluble-Bound Phenolics (IBP)

The methods established by Wang et al. [46], Arruda et al. [47], and Yao et al. [10] were employed to perform the extraction of free phenolics, soluble conjugated phenolics (esterified and glycosylated), and IBP from the peel. The sample underwent freeze-drying, utilizing a Freeze Dry System-Freezone^®^ 4.5 (Labconco Corporation, Kansas City, MO, USA), at a temperature of −45 °C ± 1 °C for 72 h. The resultant freeze-dried sample was then ground into powder by means of a stainless steel grinder (Waring blender) and passed through a 60-mesh sieve (0.30 mm) to ensure uniform particle size. Subsequently, the samples were stored at −20 °C ± 1 °C in an airtight container, in preparation for further analysis.

#### 2.8.1. Extraction of Free Phenolic

The procedure for extracting the free phenolics fraction was conducted in accordance with the methodology performed by Wang et al. [46]. Specifically, freeze-dried peel extract (0.5 g) was mixed with 10 mL of 70% methanol at a ratio of 1:20 (*w*/*v*) and the extraction process was performed in triplicate. The resulting extract mixture was homogenized in a water bath maintained at a temperature of 40 °C ± 5 °C and a speed of 150 rpm for a duration of 1 h. Subsequently, the filtrate obtained was subjected to liquid–liquid stratification thrice using 70 mL of ethyl acetate. The organic phase was identified as the ethyl acetate extract, while the remaining extract obtained from the liquid–liquid stratification was designated as the aqueous phase. The ethyl acetate extract was then obtained and redissolved in 5 mL of 50% methanol, representing the free phenolic fraction. The remaining aqueous phase extract was utilized in the extraction of soluble conjugate phenolics.

#### 2.8.2. Extraction of Soluble Conjugate Phenolic

The extraction of esterified and glycosylated phenolic fraction was performed in accordance with established methods by Wang et al. [48] and Arruda et al. [47]. The aqueous phase extract obtained from the free phenolics fraction was utilized to extract the soluble-conjugated phenolics. For the extraction of esterified phenolics, the extract underwent hydrolysis with 40 mL of 2 M NaOH for a duration of 4 hours at a temperature of 23 °C ± 1 °C and was subsequently acidified with 12 M HCl until a pH of 2.0 was achieved. The hydrolysate was then extracted thrice with 70 mL of ethyl acetate utilizing liquid–liquid stratification. The remaining extract obtained from the liquid–liquid stratification was the aqueous phase, whereas the ethyl acetate phase was regarded as the organic phase. The organic phase was dried using a rotary evaporator (Model 243668, EYELA 1L Rotary evaporator, New York, NY, USA) at a temperature of 40 °C ± 1 °C, and then dissolved in 5 mL of 50% methanol (*v*/*v*). The remaining aqueous phase extract was used in the extraction of glycosylated phenolics. The extracts were stored in an airtight container at a temperature of −20 °C ± 1 °C prior to analysis.

The extraction of glycosylated phenolics involved the use of the aqueous phase extract, following the method established by Arruda et al. [47]. The hydrolysis of the aqueous phase extract required the use of 5 mL of 6 M HCl, conducted at a temperature of 75 °C ± 5 °C and 150 rpm for 60 min. Subsequently, liquid–liquid stratification was employed to extract the hydrolysate thrice with ethyl acetate. The resulting ethyl acetate was obtained and subjected to rotary evaporator (Model 243668, EYELA 1L Rotary evaporator, New York, NY, USA) at 40 °C ± 1 °C. The extract was then redissolved in 5 mL of 50% methanol (*v*/*v*) before storage in an airtight container at −20 °C ± 1 °C for further analysis.

#### 2.8.3. Extraction of Insoluble-Bound Phenolics (IBPs)

The IBP fraction was obtained using the methods described by Wang et al. [46] and Arruda et al. [47]. The extraction of IBPs was carried out from the residue that was acquired from the free phenolics fraction. The residue was subjected to hydrolysis with 50 mL of 2 M NaOH at room temperature (22 °C ± 1 °C) for a duration of 4 h. Following this, the hydrolysate was acidified to a pH of 2.0 using 12 M HCl. The resultant mixture was then subjected to extraction thrice using 70 mL of ethyl acetate. The fractions of ethyl acetate that were obtained were subsequently subjected to drying under a rotary evaporator (Model 243668, EYELA 1L Rotary Evaporator, New York, NY, USA) at a temperature of 40 °C ± 1 °C. The IBP extract that was obtained was dissolved in 5 mL of 50% methanol (*v*/*v*) and stored in an airtight container at −20 °C ± 1 °C until analysis.

### 2.9. GC–MS Analysis of Volatile Compounds

The qualitative GC-MS analysis was carried out according to Hanafy et al. [18]. In the initial stage, MD2-PPC extracts that were obtained from the liquid extracts’ preparation were subjected to centrifugation in a 15 mL benchtop centrifuge (Hettich EBA 20 Centrifuge, Germany) at 3600 rpm for 10 minutes to obtain a clear filtrate. Thereafter, aliquots of the filtrate measuring 1.0 mL were transferred into an autosampler vial employing a chromatography syringe filter which had a pore size of 0.45 µm. For each MD2-PPC extract, a dosage of approximately 1 µL was injected in splitless mode. The column used was an HP-5MS (30 m × 250 µm × 0.25 µm in thickness) into the GC Agilent Technologies model 7890A/MS-5975C, Santa Clara, CA, USA equipped with an Agilent 5975 inert mass selective detector. Peak detection was carried out using electron ionization at 70 eV and pure helium gas (99.995%) was used as the carrier gas with a flow rate of 1 mL/min. Temperature ramping was executed from 70 °C to 150 °C at a rate of 5 °C/min with a holding time of approximately 5 min. Finally, the temperature was increased to 300 °C at 15 °C/min. The compounds that were present in each MD2-PPC extract were expressed as percentage peak areas identified based on retention time and spectral matching to data from standards (National Institute Standards and Technology (NIST) Mass Spectral 11 library) with a similarity index of >80%. The mass spectrum of extract components matched the mass spectrum fragmentation patterns stored in the library and published literature.

### 2.10. Statistical Analyses

All the data from analyses were statistically analyzed using Minitab Software (Minitab 16.0 for Windows, Minitab Ltd., Coventry, UK). A one-way analysis of variance using Tukey’s test was used for the comparison of means at a level significance of 0.05 (*p* < 0.05).

## 3. Results and Discussion

### 3.1. Composition of MD2-PPC

#### 3.1.1. Proximate Composition

The proximate composition of MD2-PPC is shown in Table 1. The moisture content of both peel and core was more than 80 g/100 g FW. Meanwhile, the protein content of the MD2 pineapple core was significantly (*p* < 0.05) higher than the peel. The presence of protein in the peel and core originated mainly from the hydroxyproline-rich glycoprotein present in the primary cell wall of the plant [48]. Glycoproteins are mainly attached to the cellulose of primary cell walls in the peel, forming a network of microfibrils [49]. Pardo et al. [50] also reported a lower protein content in Mexican pineapple peel than in the core. Furthermore, the fat content for both MD2-PPC was less than 1.0% of the dry weight, although Morais et al. [51] reported that the total lipid content of raw pineapple peel cultivated in Brazil was approximately 1.1%. Ash and carbohydrate contents were not significantly different (*p* > 0.05) between MD2-PPC. The ash content ranged between 7.31–7.81% for both peel and core. The carbohydrate and ash content of peel in MD2 pineapple was higher when compared with the varieties of Smooth Cayenne, Tainung 17, and Tainung 19 [15].

#### 3.1.2. Chemical Composition

The pH values, titratable acidity, and total soluble solids (TSS) determined from MD2-PPC are presented in Table 1. The pH values of MD2-PPC were approximately 4.00. Similar pH values of 3.63 and 4.04 were reported by Campos et al. [21] for fresh pineapple peel (purchased from Costa Rica) and by Rivera et al. [52] for dehydrated pineapple peels, respectively. The low pH values are due to the high amount of citric acid and malic acid present in the fruits [53]. Titratable acidity (% citric acid) in MD2 pineapple peel was significantly higher (*p* < 0.05) than in the core (Table 1). Similar findings were recorded for pineapple variety N36, which had higher titratable acidity (%) in the peel than the core [54]. Titratable acidity is a measurement of organic acids (% citric acid) in fruits. Therefore, high titratable acidity indicates lower pH values. The TSS content in MD2-PPC shown in this study was higher than the N36 pineapple variety as reported by Nadzirah et al. [54]. A low TSS content was also reported by Ketnawa et al. [53] for *Nang Lae* and *Phu Lae* pineapple peel and core variety, ranging between 4.33–6.27 °Bx. Therefore, the findings of MD2 pineapple peel and core compositions show that the differences in pH values and total soluble solid content are influenced by the maturity index, variety and geographical location, and condition of the pineapples [55].

#### 3.1.3. Sweetness Index (SI) and Astringency Index (AI)

The findings of SI and AI were displayed in Table 1. A statistically significant difference (*p* < 0.05) was observed between peel and core. The SI for the MD2 pineapple core was twice as high for the peel. The presence of phytochemicals and aroma compounds in MD2 pineapple peel and core contribute to astringency. The AI was shown to be significantly (*p* < 0.05) higher in MD2-PPC, which was due to the interaction of the high total phenolic and flavonoid (phytochemicals) contents when bonded with salivary peptides. The astringency and mouthfeel of food products such as tea and wine have been associated with the interaction of phytochemicals and peptide binding produced by proline-rich salivary proteins [56].

### 3.2. TPC, TFC, and DPPH Free Radical Scavenging Activity

Antioxidant activities in fruits are indicated by the presence of phenolics such as phenolic acids and flavonoids [57] and were measured by DPPH and IC_50_ as shown in Table 2. The peel demonstrated a significant difference (*p* < 0.05) in TPC and TFC in comparison to the core, exhibiting a 47% and 108% rise, respectively (Table 2). Moreover, TPC was more abundant than TFC in MD2-PPC. The MD2 pineapple peel extract showed a greater potential for DPPH free radical scavenging of 53% (1 mg/mL) compared with the core extract. As shown in this study, the peel had the ability to scavenge DPPH free radicals (IC_50_ = 0.63 mg/mL), but in comparison, the core extract could not achieve 50% DPPH inhibition. In other studies, the IC_50_ of Bali variety pineapple peel was reported as 1.13 mg/mL [45], and four different varieties of apple peel were reported as 3–6 mg/mL [12]. Thus, peel extract of the MD2 pineapple variety had a potent natural antioxidant ability.

### 3.3. Free, Soluble Conjugate, and IBP Fractions

The phenolic fraction of MD2 pineapple peel extract is shown in Figure 2. Glycosylated phenolics showed the highest recovery of TPC, more than 100% compared with other phenolics, in the following order glycosylated > esterified > IBP > free phenolics. The findings of this study were consistent with those for pomegranate peel in which there was a higher TPC in the soluble fraction compared with IBP, as reported by Gulsunoglu et al. [58]. The various phenolic fractions associated with a plant matrix could be divided into free, soluble conjugate, and IBP fractions [9]. Glycosylated phenolics are considered as soluble conjugated phenolics that can be found covalently bound or esterified to sugar and fatty acids molecules of low molecular mass components [59]. In the solid–liquid extraction of MD2 pineapple peel, as shown in this study, soluble conjugated phenolics were the largest extract. However, the most important phenolic components with many health benefits that have garnered scientific interest are the IBPs. The extraction of IBPs from MD2 pineapple peel was 0.16 mg GAE/100 g of TPC. Therefore, this study showed that MD2 pineapple peel extract contained various phenolic fractions through a solid–liquid extraction method.

### 3.4. Volatile Compounds

Thirty-eight compounds were detected in MD2 pineapple peel and only twenty-three compounds were detected in the core (Figure 3 and Table 3). Through the qualitative screening of volatile compounds, aldehydes were the predominant aroma compounds in the MD2-PPC, comprising 47% and 36%, respectively. Other volatile constituents present in the MD2-PPC at only low amounts were ketones (8.82% and 0.4% in peel and core, respectively), alcohol (0.39% in the peel), and fatty acids (4.24% and 3.26% in peel and core, respectively).

MD2-PPC, 5-(hydroxymethyl)-furan-2-carbaldehyde (Table 3) was the most abundant compound. A finding reported by Žemlička et al. [37] found that in MD-2 pineapple fruit cultivated in Costa Rica, the aldehyde group was a minor constituent of aroma compounds, and the ester group was the major constituent. Shapla et al. [60] also reported that HMF provided therapeutic effects including antioxidant, anti-allergen, and anticarcinogen activity, and protection against sickle cell anemia and hypoxic injury.

The most abundant compound detected in the MD2-PPC was 3,5-dihydroxy-6-methyl-2,3-dihydropyran-4-one (DDMP). DDMP is a strong antioxidant compound, as identified by Yu et al. [61] in glucose–histidine Maillard reaction products. Chen et al. [62] reported DDMP as an antioxidant compound with strong free radical scavenging ability. Studies in food have also identified the presence of DDMP found in fig honey [63], and dried prunes [64]. Furthermore, Ban et al. [65] reported that DDMP isolated from onion induced apoptotic cell death to inhibit colon cancer growth. DDMP in *Tabebuia rosea* leaves has also been reported to have antimicrobial properties [66]. Hence, the MD2-PPC could exhibit the same above-mentioned properties.

The compound, which was present in both MD2-PPC extracts, was 1,2-benzenediol. In plants, 1,2-benzediol is synthesized by the shikimate pathway as a phenolic compound [67]. As reported by Mathew et al. [68], the compound 1,2-benzenediol is one of the major constituents in the stem and leaves of the Josephine hybrid pineapple variety. Kim and Lee [67] also isolated 1,2-benzenediol from persimmon roots (*D. kaki roots*), which showed antimicrobial properties against food-borne bacteria.

1,4-Benzenediol (phenolic acid), which was also found in MD2-PPC, is often applied as an agrochemical, antioxidant, pharmaceutical, and perfume compound, as well as in the rubber industry, as described by Patil et al. [69]. Wheat husk bio-oil [70] and seed oil of *Celastrus paniculatus* Willd. also contained 1,4-benzenediol [71]. The compound 1,3-benzenediol was detected in the core extract. Patil et al. [69] reported that 1,3-benzenediol is widely used for medical, antioxidant, and photographic applications, and in cosmetics, pesticides, and flavoring agents. 1,3-Benzenediol was also found in the stem and leaves of the Josephine hybrid pineapple variety [68].

Another minor component of significance was n-decanoic acid, a medium-length (10-carbon) fatty acid [72]. The fatty acid compound was found in both MD2-PPC. The compound n-decanoic acid exhibited anti-inflammatory and antibacterial properties against *Propionibacterium* acnes, which causes acne inflammation [73]. n-Hexadecanoic acid was also present in the MD2-PPC, at 1.72% and 1.95%, respectively. According to Aparna et al. [74], n-hexadecanoic acid (commonly known as palmitic acid) can function as an anti-inflammatory agent. 3,4-Dimethoxybenzoic acid was also found in the peel. The fatty acids found in both MD2-PPC also contribute to the aroma profile. Engelen and de Wijk [75] stated that small quantities of volatiles such as fatty acids do contribute to the intensification of sensory perception orally.

3,4-Dimethoxybenzoic acid is a derivative of benzoic acid that is commonly found in fruits and plants. This compound has been shown to possess anti-inflammatory, antimicrobial, and medicinal properties. Additionally, it has been observed to have a protective effect against lipopolysaccharide-induced acute lung injury in mice [76]. Saravanakumar and Raja [77] also reported that veratric acid exhibit antihypertensive and antioxidant effects in L-NAME-treated hypertensive rats.

Other minor compounds which constitute the MD2-PPC were maltol (3-Hydroxy-2-methylpyran-4-one) and L-arabinitol ((2S,4S)-Pentane-1,2,3,4,5-pentol) in peel and core, respectively. According to Kordowska-Wiater [78], arabinitol has a similar sweetness level as sucrose with a low-calorie value. In addition, maltol is an active constituent found in Passionflower (*Passiflora incarnata*) [79] and balsam fir *Abies balsamea* (L.) Mill. (Pinaceae) [80] identified to have therapeutic properties. Therefore, the presence of carbohydrate compounds in MD2-PPC would provide potential alternatives to sugars in the food industry.

## 4. Conclusions

The MD2 pineapple peel (maturity index of 2) exhibited higher TPC, TFC, and DPPH free radical scavenging ability compared with the core. Meanwhile, the extraction of different phenolic fractions showed glycosylated phenolic as the most abundant from the peel. Furthermore, the highest number of volatile compounds detected in both MD2-PPC were aldehyde (2-furan carboxaldehyde, (5-hydroxymethyl)), and 2,3-dihydro-3,5-dihydroxy-6-methyl-4H pyran-4-one (DDMP). Furthermore, this study would contribute to the emerging field of natural compound extraction from fruit peels. Therefore, the phenolics and volatile compounds identified from the extract of MD2 pineapple peel may be utilized in various industries such as food packaging, food supplements, beverages, cosmetics, and pharmaceuticals.

## Figures and Tables

**Figure 1 foods-12-02233-f001:**
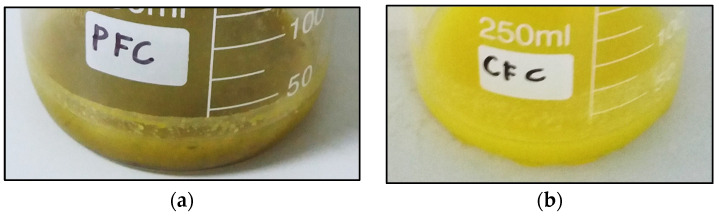
The extracts of MD2 pineapple (**a**) peel and (**b**) core.

**Figure 2 foods-12-02233-f002:**
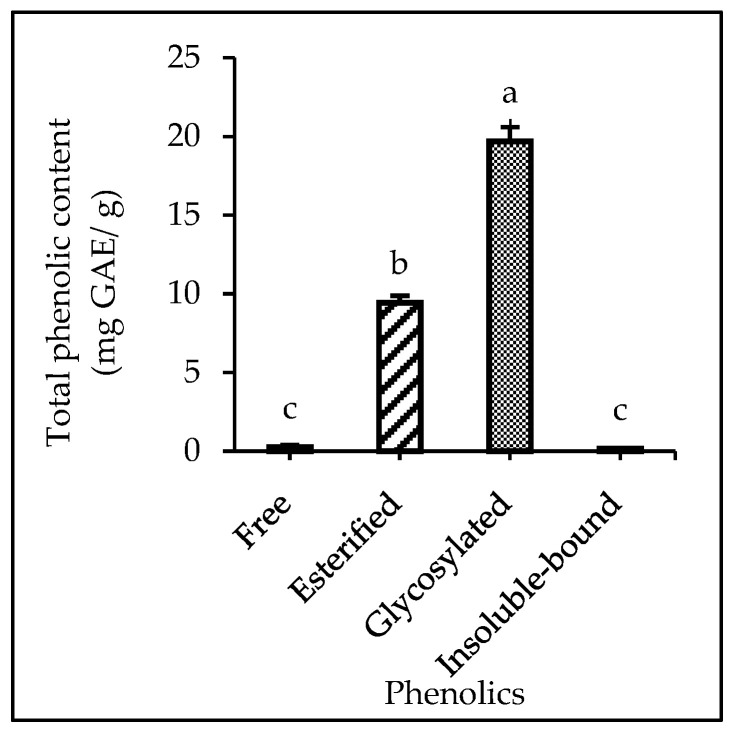
The different phenolic fractions (free, esterified, glycosylated, and IBPs) extracted from the peel of MD-2 pineapple. The different letters in the same row indicate the values are significantly different (*p* ≤ 0.05).

**Figure 3 foods-12-02233-f003:**
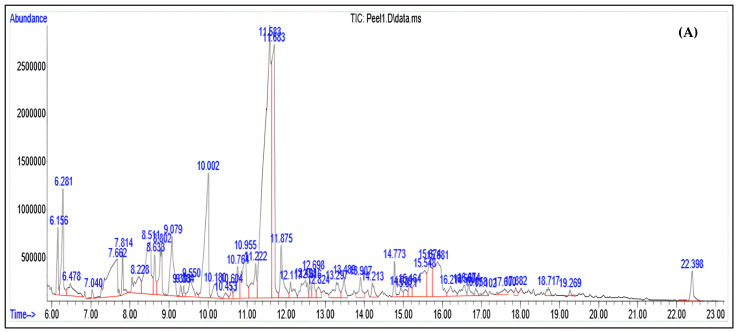
Chromatograms of volatile compounds found in MD2 pineapple (**A**) peel; (**B**) core.

**Table 1 foods-12-02233-t001:** Composition of MD2-PPC.

Parameter	Peel	Core
**Proximate composition**		
Moisture (g/100 g FW)	85.14 ± 1.37 ^a^	86.90 ± 1.34 ^a^
Protein (g/100 g FW)	0.58 ± 0.00 ^b^	1.25 ± 0.17 ^a^
Fat (g/100 g FW)	0.75 ± 0.20 ^a^	0.38 ± 0.09 ^b^
Ash (g/100 g FW)	7.13 ± 0.54 ^a^	7.71 ± 0.54 ^a^
Total carbohydrate (g/100 g FW)	6.40 ± 1.82 ^a^	3.76 ± 2.47 ^a^
**Chemical composition**		
pH	4.00 ± 0.04 ^a^	3.96 ± 0.03 ^a^
Titratable acidity (% citric acid)	0.74 ± 0.08 ^a^	0.32 ± 0.02 ^b^
Total soluble solids (ºBrix)	9.34 ± 1.53 ^a^	12.00 ± 1.00 ^a^
Sweetness index (SI)	12.84 ± 3.11 ^a^	37.66 ± 4.61 ^b^
Astringency index (AI)	0.08 ± 0.02 ^a^	0.03 ± 0.01 ^b^

The result is based on g/100 g of fresh weight sample. Values represent the mean ± SD (*n* = 3). The different letters in the same row indicate the values are significantly different (*p* ≤ 0.05).

**Table 2 foods-12-02233-t002:** Total phenolic content, total flavonoid content, and IC_50_ values of MD2-PPC.

Analysis	Peel	Core
Total Phenolic Content (mg GAE/100 g FW)	49.35 ± 1.28 ^a^	33.50 ± 0.24 ^b^
Total Flavonoid Content (mg QE/100 g FW)	12.80 ± 1.84 ^a^	6.15 ± 1.12 ^b^
IC_50_ (mg/mL)	0.63 ± 0.01	^1^ NA

GAE: gallic acid equivalent; QE: quercetin equivalent; FW: fresh weight; ^1^ NA: IC_50_ was not determined with up to 0.50 mg/mL. Values represent the mean ± SD (*n* = 3). The different letters in the same row indicate the values are significantly different (*p* ≤ 0.05).

**Table 3 foods-12-02233-t003:** Volatile compounds identified in MD2-PPC extracts.

Peak No.	RT (Min)	Compound Name ^a^	Nature of Compound	Molecular Formula	Percentage Peak Area (%)	
					Peel	Core
1	5.38	Furan-2-ylmethanol	Alcohol	C_5_H_6_O_2_	0.39	ND
2	5.84	Methyl (*Z*)-*N*-hydroxybenzenecarboximidate	Imines	C_8_H_9_NO_2_	0.22	ND
3	6.16	2(5H)-Furanone	Ketone	C_4_H_4_O_2_	2.51	ND
4	6.28	Cyclopentane-1,2-dione	Ketone	C_5_H_6_O_2_	4.52	ND
5	7.03	3,5-Dihydroxy-6-methyl-2,3-dihydropyran-4-one (DDMP)	Flavonoid ^b^	C_6_H_8_O_4_	0.29	ND
6	7.28	3H-Pyran-2,6-dione	Flavonoid ^b^	C_5_H_4_O_3_	0.44	ND
7	7.81	2-Hydroxy-3-methylcyclopent-2-en-1-one	Ketone	C_12_H_16_O_4_	1.46	ND
8	7.88	(2S,4S)-Pentane-1,2,3,4,5-pentol	Carbohydrate	C_5_H_12_O_5_	ND	0.72
9	8.63	Furan-2,5-dicarbaldehyde	Aldehyde	C_6_H_4_O_3_	0.89	ND
10	8.78	Methyl furan-3-carboxylate	Carboxylic acid	C_6_H_6_O_3_	0.98	ND
11	9.04	Carbamic acid, (2-hydroxy-1-methyl ethyl)-, 1,1-dimethyl ethyl ester, (s)-	Carboxylic acid	CH_3_NO_2_	ND	3.16
12	9.08	N’-methylpropane-1,3-diamine	Diamine	C_4_H_12_N_2_	5.62	ND
13	9.30	3-Hydroxy-2-methylpyran-4-one	Carbohydrate	C_6_H_6_O_3_	0.42	ND
14	9.38	3-Ethyl-2-hydroxycyclopent-2-en-1-one	Ketone	C_7_H_10_O_2_	0.33	ND
15	9.54	2-Propyltetrahydropyran	Oxime	C_8_H_16_O	1.47	ND
16	9.70	2,6-Dimethylpyran-4-one	Flavonoid ^b^	C_7_H_8_O_2_	0.07	ND
17	9.99	3,5-Dihydroxy-6-methyl-2,3-dihydropyran-4-one (DDMP)	Flavonoid ^b^	C_6_H_8_O_4_	ND	3.32
18	10.01	3,5-Dihydroxy-6-methyl-2,3-dihydropyran-4-one (DDMP) (I)	Flavonoid ^b^	C_6_H_8_O_4_	12.67	ND
19	10.17	Benzoic acid	Phenolic acid ^c^	C_6_H_5_COOH	1.67	ND
20	10.45	(2-Hydroxyphenyl) urea	Urea	C_7_H_8_N_2_O_2_	0.23	ND
21	10.48	4-Hydroxyoxolan-2-one	Ketone	C_4_H_6_O_3_	ND	0.4
22	10.76	Benzene-1,2-diol	Phenol ^c^	C_12_H_12_O_3_	1.43	1.64
23	10.85	5-Methylfuran-2-carbaldehyde	Aldehyde	C_6_H_6_O_2_	0.67	ND
24	11.22	Malonic acid, di(3-methylpent-2-yl) ester	Dicarboxylic acid	C_15_H_28_O_4_	0.61	ND
25	11.44	5-Methylfuran-2-carbaldehyde	Aldehyde	C_6_H_6_O_2_	ND	35.73
26	11.58	5-(Hydroxymethyl) furan-2-carbaldehyde	Aldehyde	C_6_H_6_O_3_	44.7	ND
27	11.83	1,2-Benzenediol, 3-methyl-	Phenol ^c^	C_7_H_8_O_2_	ND	2.14
28	11.87	3-Methylbenzene-1,2-diol	Phenol ^c^	C_7_H_8_O_2_	2.94	ND
29	12.09	Benzenethiol, o-isopropyl	Aroma	C_9_H_12_S	ND	0.34
30	12.11	2-Propan-2-ylbenzenethiol	Aroma	C_9_H_12_S	0.32	ND
31	12.14	Benzene-1,4-diol	Phenol ^c^	C_12_H_16_O_2_	ND	0.16
32	12.34	4-Methylbenzene-1,2-diol	Phenol ^c^	C_7_H_8_O_2_	0.19	ND
33	12.37	4-Methylbenzene-1,2-diol	Phenol ^c^	C_7_H_8_O_2_	ND	1.26
34	12.40	4-Methylbenzene-1,2-diol	Phenol ^c^	C_7_H_8_O_2_	0.26	ND
35	12.61	(5-Formylfuran-2-yl) methyl acetate	Furfural	C_8_H_8_O_4_	0.44	ND
36 37	12.70 13.25	4-Ethenyl-2-methoxyphenol 2-Methylbenzene-1,4-diol **(V)**	Phenol ^c^ Phenol ^c^	C_9_H_10_O_2_ C_7_H_8_O_2_	0.84 ND	ND 0.51
38	13.29	2-Methylbenzene-1,4-diol **(V)**	Phenol ^c^	C_7_H_8_O_2_	0.29	ND
39	13.32	2,6-Dimethoxyphenol	Phenol ^c^	C_8_H_10_O_3_	ND	0.5
40	13.49	n-Decanoic acid **(4)**	Fatty acid	C_10_H_20_O_2_	0.98	0.9
41	13.84	4-Ethylbenzene-1,3-diol **(VI)**	Phenol ^c^	C_8_H_10_O_2_	ND	0.56
42	13.90	2-Fluoro-1,3,5-trimethylbenzene	Fluorobenzene	C_9_H_11_F	0.77	ND
43	14.77	Methyl (2E,4E)-octa-2,4-dienoate **(3)**	Fatty acid	C_9_H_14_O_2_	1.31	0.64
44	14.91	2,6-Dimethylbenzene-1,4-diol	Phenol ^c^	C_8_H_10_O_2_	ND	0.33
45	14.94	2,3-Dimethylbenzene-1,4-diol	Phenol ^c^	C_8_H_10_O_2_	0.31	ND
46	15.15	1-Fluoro-1-hex-1-ynyl-2,2-dimethylcyclopropane	Cycloalkane	C_11_H_17_F	0.23	ND
47	16.66	5-(2-Methoxyethyl)-1,2,3,4-tetramethylcyclopenta-1,3-diene	Alkadienes	C_12_H_20_O	ND	0.71
48	16.67	6-Hydroxy-4-methoxy-2,3-dimethylbenzaldehyde-	Aldehyde	C_10_H_12_O_3_	0.53	ND
49	16.86	5,6-Dimethoxy-2,3-dihydroinden-1-one	Phenol ^c^	C_11_H_12_O_3_	0.42	ND
50	16.88	Benzenethiol, 4-(1,1-dimethyl ethyl)-	Aroma	C_11_H_16_S	ND	0.18
51	18.33	3,4-Dimethoxybenzoic acid	Phenolic acid ^c^	C_9_H_10_O_4_ C_9_H_10_O_4_	0.19	ND
52	18.67	2-Amino-5-ethyl-3-nitro-benzoic acid	Phenolic acid ^c^	C_9_H_10_N_2_O_4_	0.12	ND
53	18.71	[(E)-[(E)-4-(2,6,6-Trimethylcyclohexen-1-yl)but-3-en-2-ylidene]amino]urea	Enone	C_14_H_23_N_3_O	ND	0.24
54	19.61	1-(4-Methoxyphenyl)butane-1,3-dione	Hydroxyketone	C_11_H_12_O_3_	ND	0.09
55	22.40	n-Hexadecanoic acid **(2)**	Fatty acid	C_16_H_32_O_2_	1.95	1.72

ND represents not detected. RT represents retention time. Subscripts “^a^” represents compounds identified with more than 80% similarity index compared to the standard mass spectra in NIST 11 library. Subscripts “^b^” represents the total percentage peaks of total flavonoid group at 13.47% and 3.32% in peel, respectively. Subscripts “^c^” represents the total percentage peaks of phenolic acid group at 8.66% and 7.10% phenolic acids group in core, repsectively.

## Data Availability

Not applicable.

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
