# Peer review of "Comparison of Phenolic and Volatile Compounds in MD2 Pineapple Peel and Core"

_foods, 2023, doi:10.3390/foods12112233_

Round 1

Reviewer 1 Report

the paper is interesting, althogh some improvements are needed:

-could you in section 2.9 describe preparation of extract for gas chromatography

-in line 318 use M instead of N (normality)

-can you put pictures of extracts?

Reviewer 2 Report

Please find my comments below:

·       The article presented for review is interesting, but for me underdeveloped.

·       First, the introduction is chaotic and overly elaborate. I think it should be shortened.

·       I wonder what really was the main purpose of the research. Was it the desire to find out what is in the pineapple peel and core or indeed the desire to manage the waste. If it's the latter, I miss the answer as to how this would be feasible. None of the processes have a calculated yield.  Besides, I wonder if it would be economically beneficial to extract substances from the waste.

·       Please combine the chapters Results, Discussion of results and Discussion.

·       Table 3 lacks the calculated retention times for volatile compounds as well as their comparison with the literature. Besides, why were terpene standards not used to determine retention indices? Probability-based determination of substances is not sufficient.

·       In my opinion, the conclusions chapter should be rewritten.

·       All Latin names of microorganisms and plants are written in italics.

·       The formulas of chemical compounds should be improved in the methodology.

Reviewer 3 Report

A major revision is required for the manuscript as shown below:

1.       A statement for the novelty of the study must be provided as many studies on the chemical characteristics of pineapple wastes have been published in recent decades.

2.       The discussion of the Introduction is too long. This must be concise and directly focus on the man objects of the study

3.       Line 169: please provide method for obtaining the “defatted samples”

4.       Equation 3: the method for determining ash content must be provided

5.       The DPPH radical scavenging activity of the extract must be shown as IC50 instead of the inhibition percentage. The % inhibition cannot help comparing the antioxidant power of the samples compared to other similar materials.  Actually, this figure can be removed as IC50 is also mentioned in Table 2.

6.       Notes explaining the meaning of letters (a, b, c, etc.) showing in the Figures must be provided.

7.       The brief description of the results (which appear in the “Discussion section”) must be presented in the “Results section”. Eg. Contents in lines 395-398, 411-413, 430-433, 441-449, etc.

8.       The discussion of the flavonoid and phenolic compounds based on the result of GC-MS analysis is not reasonable as this cannot exactly indicate the flavonoid and phenolic profiles of the extracts. It is well-known that most of flavonoid and phenolic compounds in the plant extracts are non-volatile substances so those are commonly quantified by LC instead by GC. Thus the concluding statements based on the result of GC-MS analysis for flavonoid and phenolic compounds are not reliable.

9.       The words “MD2 pineapple peel and core” should be abbreviated as this is repeated many times in the manuscript

Round 2

Reviewer 1 Report

Thanks for answers and corrections, please check once again english editing and literature

Author Response

Dear Editor,

The manuscript entitled “Comparison of phenolic and volatile compounds in MD2 pineapple peel and core” was revised in accordance with the constructive and insightful comments of the respected reviewers. The authors hope that the revisions made improvements in the manuscript and are suitable for publication in Foods

Reviewer 2 Report

Unfortunately, the authors did not follow my comments.

So, once again, I am asking you to change the section on results and discussion of results. Perhaps to make it easier, let the authors look at how other publications in the journal Foods are edited.

I still have reservations about the GC analysis. Comparing only the mass spectra with the NIST database is insufficient to identify the compounds!!! Please follow my previous comments!

Reviewer 3 Report

All comments/suggestions have been considered and sufficiently revised.

An acceptance for publishing is suggested for the manuscript. 

Author Response

(The authors gave the same response as above.)
